# Recovering from COVID-19 (ReCOV): Feasibility of an Allied-Health-Led Multidisciplinary Outpatient Rehabilitation Service for People with Long COVID

**DOI:** 10.3390/ijerph21070958

**Published:** 2024-07-22

**Authors:** Aruska N. D’Souza, Myvanwy Merrett, Hilda Griffin, An Tran-Duy, Carly Struck, Timothy N. Fazio, Genevieve Juj, Catherine L. Granger, Casey L. Peiris

**Affiliations:** 1Allied Health, The Royal Melbourne Hospital, Parkvillle, VIC 3050, Australia; 2Centre for Health Policy, Melbourne School of Population and Global Health, The University of Melbourne, Parkville, VIC 3010, Australia; 3Methods and Implementation Support for Clinical and Health Research (MISCH) Hub, Faculty of Medicine, Dentistry and Health Sciences, The University of Melbourne, Parkville, VIC 3010, Australia; 4Health Intelligence Unit, The Royal Melbourne Hospital, Parkvillle, VIC 3050, Australia; 5Electronic Medical Records, The Royal Melbourne Hospital, Parkvillle, VIC 3050, Australia; 6Melbourne Medical School, The University of Melbourne, Parkvillle, VIC 3010, Australia; 7School of Allied Health, Human Services and Sport, La Trobe University, Melbourne, VIC 3086, Australia

**Keywords:** post-acute COVID-19 syndrome, outpatients, feasibility studies

## Abstract

Background: A multidisciplinary approach is required for the management of long COVID. The aim of this study was to determine the feasibility (demand, implementation, practicality, acceptability, and limited efficacy) of an allied-health-led multidisciplinary symptom management service (ReCOV) for long COVID. Methods: A single-group observational cohort feasibility study was conducted to determine demand (referrals), acceptability (survey), implementation (waitlist times, health professions seen), practicality (adverse events), and limited efficacy (admission and discharge scores from the World Health Organization Disability Assessment Scale, Brief Illness Perception Questionnaire (BIPQ), Patient Health Questionnaire, and EuroQol 5D-5L). Data are presented as median [interquartile range] or count (percentage). Results: During the study, 143 participants (aged 42.00 [32.00–51.00] years, 68% women) participated in ReCOV. Participants were waitlisted for 3.86 [2.14–9.86] weeks and engaged with 5.00 [3.00–6.00] different health professionals. No adverse events occurred. The thematic analysis revealed that ReCOV was helpful but did not fully meet the needs of all participants. Limited efficacy testing indicated that participants had improved understanding and control (*p* < 0.001) of symptoms (BIPQ) and a small improvement in EQ VAS score (median difference 5.50 points [0.00–25.00], *p* = 0.004]). Conclusions: A multidisciplinary service was safe and mostly acceptable to participants for the management of long COVID. Further research should investigate the clinical and cost effectiveness of such a service, including optimal service duration and patient outcomes.

## 1. Introduction

It is estimated that 7–17% of people with COVID-19 develop long COVID, and, globally, at least 65 million people are thought to have experienced this condition [1,2,3]. Long COVID, also described as post-COVID-19 condition or post-acute sequelae of COVID-19, is broadly defined as persistent or new COVID-19 symptoms for at least 12 weeks after the initial infection, with symptoms lasting for at least two months without another explanation [1,4]. Over 200 different symptoms of long COVID have been reported, including fatigue, shortness of breath, cough, chest pain, nausea, disordered sleep, memory loss, and difficulty concentrating [1,5,6]. It has an impact on numerous organs and body systems including the heart, blood vessels, lungs, immune system, gastrointestinal tract, and neurological system [1]. The hypothesised biological mechanisms for this include neuroinflammation, viral persistence, autoimmunity, and excessive blood clotting, so more research is required to confirm these hypotheses [1]. Long COVID has a substantial impact on health-related quality of life (HRQoL) and may increase disability, which also impacts participation in society, including employment [1,7,8,9]. It is anticipated that long COVID will have significant social, healthcare, and economic impacts [2,10]; consequently, effective management is needed.

Given the diverse nature of symptoms, it is recommended that multidisciplinary healthcare services and resources are utilised to support the needs of those with long COVID [11]. Moreover, as the exact pathophysiological mechanism of long COVID is not fully understood, there are limited effective treatments, but strategies to support symptom management are recommended as fundamental [1,11,12]. At the time of this study, there were limited services dedicated to the management of long COVID and a paucity of evidence-based data on the efficacy of these services [11,12,13]. Recently, a feasibility study (n = 30) observed that 50% of participants with long COVID were symptom-free after attending outpatient multidisciplinary rehabilitation, though the study duration was not specified [14]. Participants also demonstrated significant improvements in strength, perceived physical and mental health (including depression and anxiety), and cardio pulmonary parameters [14]. In April 2022, the Royal Melbourne Hospital established ReCOV, an allied-health-led clinic, to provide a multidisciplinary symptom management approach for patients with long COVID. The aim of this study was to determine the feasibility of ReCOV with respect to acceptability, demand, implementation, practicality, and limited efficacy.

## 2. Materials and Methods

### 2.1. Design

An observational, single-site cohort feasibility study was conducted using a feasibility evaluation framework [15]. The feasibility domains of acceptability, demand, implementation, practicality, and limited efficacy were evaluated [15]. The ReCOV service was described according to the telehealth extension of the Template for Intervention Description and Replication—Telehealth (TIDieR-Telehealth) checklist and guide (Appendix A) [16]. Ethics approval (HREC/89125/MH-2022) was received from the hospital’s Human Research Ethics Committee (HREC), and participants provided consent to participate. 

### 2.2. Participants and Setting

This study was conducted at a large tertiary metropolitan hospital in Melbourne, Australia, from May to December 2022. Adults (18+ years) who were more than four weeks after COVID-19 infection with ongoing symptoms were eligible for the ReCOV service and were included in this study if they attended. While long COVID is characterised as ongoing symptoms for at least 12 weeks, 4 weeks was selected to allow for potential time on a waitlist [11,13]. Participants were referred to the ReCOV service by their general practitioner (GP), an outpatient service, ward, or emergency department of the hospital, or via self-referral if they were staff. Those whose symptoms pre-dated their COVID-19 infection or were not primarily caused by their COVID-19 infection or those with underlying chronic health conditions that were exacerbated by COVID-19 were excluded. 

### 2.3. Intervention 

The ReCOV allied-health-led service was modelled on Austin Health’s Long COVID service (another Victorian healthcare provider), the available literature at the time [10,11,12,13], clinical knowledge, and expertise of chronic disease management (including the Victorian Post COVID-19 Research Group, physicians, and community partners). Participants were emailed a triage survey to guide referral to health professionals within the service. The survey also included standardised and validated outcome measures regarding symptoms, general health, quality of life, anxiety and depression, dyspnoea, and malnutrition risk (Appendix A). Participants completed this electronic triage survey either independently or via a phone call with ReCOV administration staff and a phone interpreter (if required). An allied health assistant reviewed responses and referred participants to relevant health professionals based on reported symptoms and clinical indicators (Table 1). Available professions included physiotherapy, occupational therapy, exercise physiology, social work, clinical psychology, neuropsychology, dietetics, rehabilitation physician, and music therapy. It was planned that all appointments were delivered within 12 weeks, but due to the wait listing time for certain professions, participants were subsequently offered 12 weeks per profession. This tailored triage along with fortnightly multidisciplinary meetings enabled the coordination of patient-centred care. The electronic survey was repeated on discharge from the ReCOV service, and reminders were sent up to three times at both timepoints.

Allied health interventions were provided based on participant-identified goals and health-related outcome measures and are described in Table 1 and Appendix A. Appointments were mostly provided via telehealth, but face-to-face appointments were possible at the health professional’s discretion (for example, vestibular assessment or the provision of an exercise program were usually completed in person). Interventions were largely delivered one-on-one and small group sessions were offered for peer support. 

### 2.4. Outcomes

The primary outcomes were the feasibility domains of acceptability, demand, implementation, practicality, and limited efficacy [15]. Table 2 outlines the data sources and feasibility domains. Demand for the service was established by recording referral and participant numbers (including demographic details) and completion rates. Implementation was evaluated by calculating waitlist times, time from admission to discharge within the service, the number of sessions, and the number of different health professionals accessed within the ReCOV service. Practicality was assessed by analysing participant comments on ReCOV delivery and recording the number of adverse events (for example, a fall during exercise) using usual hospital reporting systems. Acceptability was determined by analysing free-text comments in the discharge survey, where participants were asked to provide written feedback about the ReCOV service. In the absence of COVID-19-specific validated outcome measures at the time of ReCOV development, outcome measures were selected for limited efficacy testing if they were validated and had published normative data related to the broad range of long COVID symptoms. These outcomes were collected via electronic survey on admission and discharge and included the World Health Organization Disability Assessment Scale (WHODAS 2.0) [17,18], questions from the Brief Illness Perception Questionnaire (BIPQ) [19], the Patient Health Questionnaire (PHQ4) [20,21], and the EQ-5D-5L questionnaire [22,23]. The World Health Organization Disability Assessment Scale 2.0 (WHODAS 2.0) comprises 36 questions in 6 domains (cognition, mobility, self-care, getting along with people, life activities, and participation) to measure health and disability. The resultant summary score ranges from 0 (no disability) to 100 (full disability), and it has been found to be a valid tool for detecting change after intervention [17,18]. Three questions from the Brief Illness Perception Questionnaire (BIPQ) were used to measure participants’ cognitive and emotional representations of their experience of symptoms, control over their condition, and understanding of their condition on a 10-point Likert scale [19]. The 4-item Patient Health Questionnaire (PHQ4) is a brief measure of anxiety and depression using a 4-point Likert scale [20,21]. Higher scores on the PHQ4 are associated with increased functional impairment and healthcare use related to anxiety and depression [20,21]. The EQ-5D-5L questionnaire was used at admission and discharge to assess patients’ self-rated health on a vertical visual analogue scale (0–100) using the EQ VAS component and to evaluate health-related quality of life (HRQoL) in the domains of mobility, self-care, usual activities, pain or discomfort, and anxiety and depression using the EQ-5D descriptive system comprising five levels for each domain [22,23]. The Australian value set [24] was used to convert the response of each participant to the EQ-5D-5L questionnaire into a single number representing the preference-weighted HRQoL value, referred to as health utility, on a scale from 0 (indicating death) to 1 (indicating perfect health). 

### 2.5. Sample Size and Data Analysis

Sample size was determined by the number of people who accessed and completed the ReCOV service from May to December 2022. This time period was chosen as feasibility data were needed to inform ongoing program needs and sustainability in 2023. Data were analysed using IBM SPSS Statistics for Windows, Version 28 (IBM Corp, Armonk, NY, USA). Descriptive data are presented as frequency and percentage (for categorical data) and as median and interquartile range [IQR] for continuous data as most variables were not normally distributed. As the distributions of the EQ-5D-5L health utility values were highly skewed to the left, mean and 95% confidence intervals for the health utility values at admission and discharge and their difference were computed using bootstrap with the bias corrected and the accelerated method [25]. Limited efficacy was analysed via the Wilcoxon signed ranks test for non-normally distributed data. 

Free-text comments were explored using inductive thematic analysis via a six-step process [26,27]. This was independently completed by two authors (MM and HG), both of whom are allied health clinicians (physiotherapist and dietitian, respectively) new to qualitative research and were mentored by authors experienced in qualitative research (AND and CLP). Of note, MM was involved with delivering physiotherapy within the ReCOV service. The steps taken included familiarisation with the data by reading and re-reading comments, selecting key phrases and words, assigning codes to capture data’s significance, identifying patterns in the codes, creating themes, and then reviewing themes together to come up with the final themes. 

## 3. Results

### 3.1. Outcomes

#### 3.1.1. Participants and Demand

From May to December 2022, 285 people were referred to the ReCOV service, and of the 148 that commenced, 143 (97%) completed ReCOV within the study timeframe (Figure 1). The most common reason for not accessing the service after referral was not being able to be contacted (n = 86), followed by symptoms resolving (n = 21). The 143 included participants had a median age of 42.00 [32.00–51.00] years; 68% (n = 97) were women (Table 3) and were a median of 35 [27–48] weeks after their initial COVID-19 diagnosis. The most commonly reported symptoms were changes in cognition (n = 132/134, 99%) and fatigue (n = 133/136, 83%).

#### 3.1.2. Implementation

After triage, participants were waitlisted for a median of 3.86 [2.14–9.86] weeks. Despite ReCOV being planned as a 12-week service, participants were seen for a median of 24.00 [16.00–34.00] sessions over a median of 17.00 [10.86–23.00] weeks. This was due to delays in commencement for some professionals and participant needs. Participants engaged with a median of 5.00 [3.00–6.00] different health professionals (Table 4). The most commonly accessed professions were exercise physiology (n = 113 participants, 79%), occupational therapy (n = 97, 68%), and neuropsychology (n = 95, 66%) (Table 4).

#### 3.1.3. Practicality

There were no adverse events recorded using the standard hospital procedures. Thematic analysis of the survey comments identified three themes, which are described below. The first theme suggested that the structure and duration of the service lacked clarity. Many participants reported that they were not aware of the short-term nature or structure of the service.

Participant 27: *“It would have been helpful to have more info about the service delivery model as I didn’t fully understand that it was a short-term service.”*

Participant 60: *“Whilst I understand there is only so much you can help with, this disease is so unpredictable and sometimes 12 weeks is not enough.”*

While participants valued having access to a multidisciplinary team, some preferred a more centralised approach to care, such as a support coordinator or case manager to help manage appointments.

Participant 50: *“There needs to be some sort of case manager so that patients with long COVID are not expected to call and speak to so many independent people.”*

Participants also reported a lack of communication on discharge and referral for ongoing support.

Participant 67: *“I am shocked to be discharged without consultation or communication, at a point where I am experiencing severe symptoms.”*

Some participants appreciated having access to telehealth appointments, but others found this electronic communication and telehealth more difficult to navigate (theme two).

Participant 70: *“It was great to have all appointments online.”*

Participant 65: *“Links to video calls were difficult to locate.”*

#### 3.1.4. Acceptability

The survey comments regarding thematic analysis indicated that ReCOV was helpful in the management of long COVID symptoms but did not fully meet the needs of all participants (theme three). For example, many participants reported feeling reassured and validated through the ReCOV service and were able to better manage their symptoms.

Participant 14: *“It was helpful to understand what was happening to my body, and the practical advice really helped my recovery.”*

Participant 28: *I felt understood and got treated for my symptoms.*

However, some participants reported that they did not benefit from the ReCOV service, particularly when symptoms were more severe. Many acknowledged that this may have been due to the short-term nature of the service and the limited treatments available for long COVID.

Participant 40: *“Staff were generally good and caring but did not know what to do when my condition deteriorated significantly…. There is a gap in appropriate support for those with very limited functional capacity.”*

#### 3.1.5. Limited Efficacy

Thirty-nine (27%) completed the discharge limited efficacy testing survey. Following ReCOV, participants reported some improvement in EQ VAS scores (median difference 5.5 [IQR 0–25.5]) and symptoms (Figure 2), but no change in overall disability on the WHODAS 2.0 (median difference 0 [0–0]) was observed (Table 5). Moreover, mean health utility values at discharge were higher than those at admission, although the difference in health utility values was not significant (Table 5). Similarly, participants observed no change in their symptoms but reported improvements in their overall understanding and control of their illness as per the BIPQ (Table 5).

## 4. Discussion

This feasibility study of the time-limited, allied-health-led ReCOV service described the demand for the multidisciplinary service and found it was safe and acceptable to participants. Participants reported improved control and understanding of their illness, and appreciated the reassurance provided by health professionals. However, a number of practicality issues were identified related to the service structure and duration, and the limited efficacy testing revealed little change. Further clarity for participants regarding the structure of the service, technology platforms to streamline appointments, and a more stringent ongoing referral process on discharge may have improved the feasibility of this service. This study adds to the current literature and guidelines that highlight the importance of a targeted multidisciplinary service to address the multifaceted symptoms of long COVID [11,13,14]. Future iterations of a hybrid (telehealth and face-to-face) long COVID clinic should ensure that the model of care is understood by consumers (including the time-limited nature and limited treatment options for long COVID), technology platforms and appropriate ongoing follow-up are available, and outcome measures are long COVID-specific.

Despite aligning with best practice guidelines for the management of long COVID [11,13], our findings indicate that participants did not always demonstrate an improvement in symptoms. The management of long COVID should include self-management for symptom control through education and holistic multidisciplinary rehabilitation including physical and psychological supports [11,13,14]. Our ReCOV service utilised a multidisciplinary approach to provide education regarding long COVID and facilitated tailored goal setting but demonstrated minimal improvement. The recovery time for long COVID is variable, and people have reported symptoms after one year from their initial illness [28]; this may explain why participants may not have always improved. Moreover, the ReCOV clinic was set up rapidly in response to an acute and increasing community need based on the best available evidence and knowledge at that time, and outcome measures were selected to measure change in known common symptoms. Subsequent research has established long COVID core outcomes and a core outcome measurement set [29,30]. While there is overlap between the measures selected and these recommendations, it is possible that a selection of measures such as the Symptom Burden Questionnaire for long COVID as well as adequate follow-up may have yielded different limited efficacy testing findings [29,31], considering that we had a large amount of missing data.

Our ReCOV service utilised existing technology interfaces to deliver a novel triage system and model of care, but further refinement may be necessary before this is scaled to other clinical areas. Our survey’s acceptability responses indicated that participants did not understand the duration of this service, and some requested assistance with managing appointments. The duration of the service was determined based on the principles of management in other chronic diseases while accounting for the rising number of COVID-19 cases within the community [32]. The inclusion of information regarding the clinical structure as well as general information regarding long COVID within the initial triage questionnaire or provided by a coordinator may have improved the understanding of the service. A transdisciplinary model of care (where a healthcare professional provides aspects of another profession’s care) may also have been of benefit to efficiently reduce the burden of multiple appointments [33,34]. The adequate testing of the electronic interface and appointment processes may also have improved the participants’ experience. Moreover, at the time of ReCOV service implementation, there were limited options for long COVID management in the community, particularly with a co-ordinated, multidisciplinary approach to care. In addition to implementing this service, ensuring that clinicians in the community are adequately equipped to manage this complex condition and have clear referral pathways may also help with ensuring that participants’ ongoing needs are met. Finally, balancing financial sustainability with the above considerations should be incorporated into future undertakings of similar services.

### Strengths and Limitations

This study utilised published frameworks and guidelines to evaluate the feasibility of a long COVID service. Qualitative and quantitative components were examined independently and integrated to provide information regarding contextual understanding and breadth of inquiry [35]. However, this study was conducted at a single site, had no control group, and the sample size was limited to the number of people who accessed the service. Despite multiple prompts, there were limited discharge survey data, with a response rate of only 27%. Integrating the discharge survey into the final ReCOV session may have improved the discharge survey completion rates. Outcome measures selected for limited efficacy testing were based on available evidence at the time; selection from the more recently published long COVID core outcome measure set [29,30] may be more sensitive to change. Semi-structured interviews or focus groups may also have provided a more in-depth understanding of acceptability. Further research is required to determine the clinical and cost effectiveness of such a service, especially with respect to optimal service duration and patient outcomes.

## 5. Conclusions

This study examined the feasibility of a time-limited multidisciplinary allied-health-led ReCOV service for people experiencing long COVID. The service was safe and mostly acceptable to participants. Participants reported an overall improvement in the understanding of their long COVID illness; however, not all participants had symptom improvement with respect to limited efficacy testing. This may have been due to the short-term nature of the service or due to the limited available treatment options for long COVID. Future iterations of the service should ensure that the model of care is explained to participants, outcome measures are long COVID-specific, and technology platforms and referral pathways are streamlined to improve patient acceptability.

## Figures and Tables

**Figure 1 ijerph-21-00958-f001:**
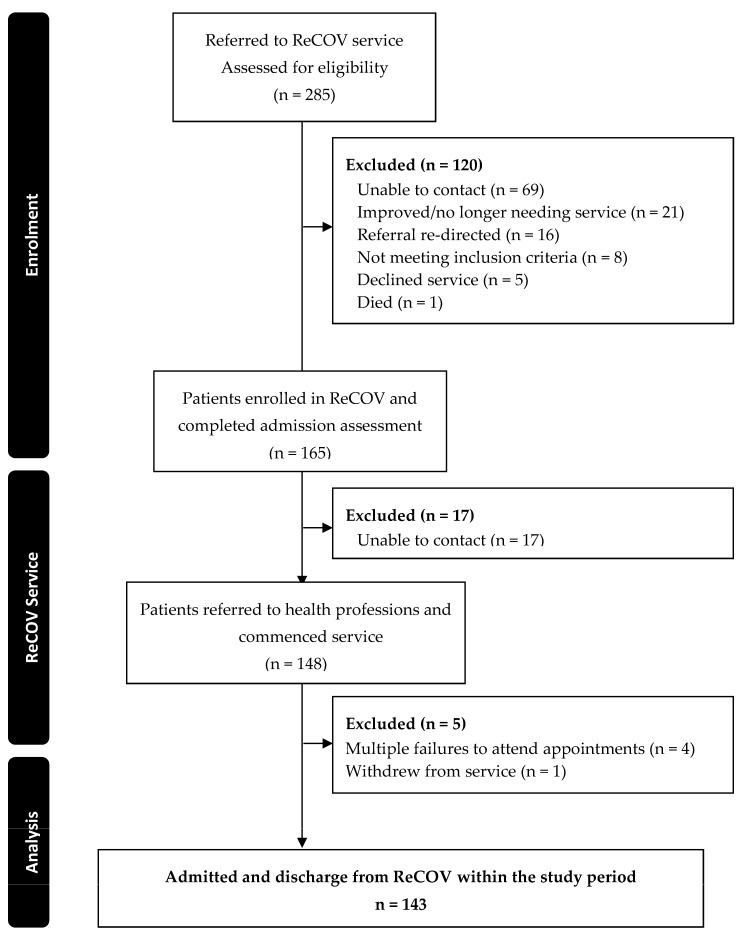
Flow of participants through this study.

**Figure 2 ijerph-21-00958-f002:**
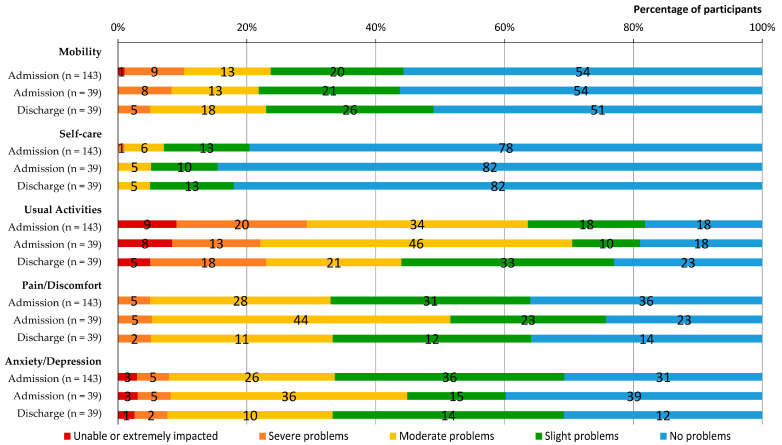
EQ-5D-5L on admission to and discharge from ReCOV.

**Table 1 ijerph-21-00958-t001:** Clinical indications and interventions provided.

Profession	Key Clinical Indication	Main Interventions Provided
Clinical Psychology	Low mood Anxiety Sleep issues	Goal-based tailored psychology sessions utilising acceptance and commitment therapy and or cognitive behavioural therapy
Dietetics	Poor appetite and malnutrition risk Weight management	Nutrition counselling and or rehabilitation for symptom management (for example, inflammatory processes, malnutrition, sarcopenia, or weight management)
Exercise Physiology	Reduced exercise tolerance Shortness of breath on exertion	Pacing and fatigue management Return to exercise (including addressing strength and balance deficits)
Music Therapy	Breathlessness Reduced breath capacity Low mood	Voice and breathing exercises for vocal fatigue and voice production Live music and relaxation for stress management Education for selection of therapeutic playlists
Neuropsychology	Reduced attention Working memory or executive dysfunction	Cognitive rehabilitation using a range of cognitive behavioural therapy-based interventions (including acceptance and commitment therapy and sensory modulation)
Occupational Therapy	Fatigue Sleep issues	Timetabling Recommendations regarding return to work 1:1 and/or group education regarding fatigue and sleep Education regarding lifestyle choices, meaningful activity, pacing, and prioritising
Physiotherapy	Dizziness Falls	Vestibular assessment and rehabilitation Education on falls prevention Strength and balance assessment and retraining
Rehabilitation Physician	Chest pain Palpitation Nausea Headaches	Recommendations regarding return to work or study Pharmacotherapy for symptom management Prescribed cardiovascular parameters for exercise prescription Referral to and liaison with other medical providers and specialties
Social Work	Financial distress Lack of social support	Education regarding housing security, material aid, grief and loss counselling, emotional support, carer support, transition support, advocacy and referrals for ongoing management

**Table 2 ijerph-21-00958-t002:** Feasibility data sources.

Feasibility Domain [15]	Data Source(s)
Demand	Participant demographics and characteristics
Implementation	Waitlist and ReCOV service duration and usage
Practicality	Adverse events and participant survey
Acceptability	Participant survey
Limited Efficacy	Change in symptoms as measured through the following: - World Health Organization Disability Assessment Scale 2.0 (WHODAS 2.0) to measure cognition, mobility, self-care, getting along with others, life activities, and participation [17,18] - Questions form the Brief Illness Perception Questionnaire to measure experience of symptoms and control and understanding of illness [19] - 4-item Patient Health Questionnaire to measure anxiety and depression [20] - EQ-5D-5L to assess health-related quality of life [22,23]

**Table 3 ijerph-21-00958-t003:** Participant demographics at time listed on waitlist to ReCOV (demand, n = 143 unless otherwise stated). Data are presented as median [interquartile range] or count (percentage).

Outcome	n (%)
Age (years), median [interquartile range]	42.00 [32.00–51.00]
Female sex	97 (68%)
First Nations persons	3 (2%)
Number of COVID-19 vaccination doses	
None	0
One	2 (1%)
Two	10 (7%)
Three	38 (27%)
Four or more	24 (17%)
Missing	69 (48%)
Symptoms reported on admission to ReCOV	
Changes in cognition (thinking more effortful, slower, or brain fog, n = 134)	123 (91%)
Fatigue	113 (79%)
Changes in sleep quality	101 (71%)
Problems with voice recovery and or breath capacity	92 (64%)
Dizziness	73 (54%)
Severe headache	57 (51%)
New weakness or sensation changes (n = 134)	63 (47%)
Breathlessness except on strenuous exertion	62 (44%)
Unable to return to previous exercise regimen (n = 134)	52 (39%)
Palpitations	52 (36%)
Chest pain	52 (36%)
Cough or difficulty managing sputum (n = 134)	45 (34%)
Unexplained weight loss	34 (25%)
Persistent nausea	32 (22%)
Vision changes	17 (13%)
Falls	17 (13%)
Job or the role performed within your job has changed as a result of COVID-19 illness	59 (41%)
COVID-19 illness has had significant financial impact (e.g., debt, unable to meet essential needs or maintain stable housing, n = 134)	36 (25%)
COVID-19 illness has impacted the ability to care for others	49 (34%)
COVID-19 illness has impacted the ability to manage their usual daily routine	47 (33%)

**Table 4 ijerph-21-00958-t004:** Service usage (n = 143).

Profession	Participants Who Used the Service	Median Number of Sessions per Participant [IQR]
Exercise physiology	113 (79%)	2 [2,3,4,5]
Occupational therapy	97 (68%)	1 [1,2]
Neuropsychology	95 (66%)	2 [1,2,3]
Music therapy	60 (42%)	3 [1,2,3,4]
Clinical psychology	59 (41%)	3 [2,3,4,5]
Rehabilitation physician	45 (32%)	1 [1]
Physiotherapy	45 (32%)	1 [1,2]
Social work	37 (26%)	1 [1,2]
Dietician	35 (25%)	1 [1,2]
Total number of professions accessed		5 [3,4,5,6]

**Table 5 ijerph-21-00958-t005:** Patient-reported outcome measures.

	Admission (n = 143)	Discharge (n = 39)	Difference	*p*-Value
EQ-5D-5L				
Visual analogue scale	49.00 [35.00–67.25]	66.00 [45.00–80.00]	5.50 [0.00–25.00]	0.004
Health utility	0.73 (0.70, 0.77)	0.80 (0.71, 0.86)	0.03 (-0.03, 0.12)	0.492
PHQ-4	5.00 [3.00–8.00]	3.00 [2.00–5.00]	1.00 [−4.00–1.00]	0.041
BIPQ (0 to 10)				
Control over illness	4.00 [2.00–6.00]	5.00 [3.00–9.00]	2 [0.25–4.78]	<0.001
Understanding of illness	5.00 [3.00–7.00]	7.00 [4.00–8.00]	1.00 [0.00–5.00]	<0.001
Experience of symptoms	7.00 [5.00–8.00]	5.00 [2.00–7.00]	0.00 [−4.00–0.00]	0.001
WHODAS 2.0 (0–100%)				
Communication (%)	33.33 [16.67–45.83]	20.83 [8.33–41.67]	0.00 [−8.00–17.00]	0.169
Getting around (%)	25.00 [10.00–50.00]	10.00 [0.00–50.00]	0.00 [−5.00–10.00]	0.433
Self-care (%)	0.00 [0.00–25.00]	0.00 [0.00–6.25]	0.00 [0.00–6.00]	0.644
Getting along (%)	15.00 [0.00–35.00]	10.00 [0.00–35.00]	0.00 [−5.00–10.00]	0.896
Life activities (%)	40.63 [21.88–59.38]	28.13 [9.38–50.00]	0.00 [−3.00–16.00]	0.221
Participation (%)	34.38 [21.88–59.38]	34.38 [12.50–62.50]	0.00 [−13.00–19.00]	0.886
Overall disability (%)	28.19 [16.81–42.01]	17.12 [7.40–37.15]	0.00 [−5.84–13.64]	0.530

## Data Availability

Due to the nature of this research and ethical restrictions, supporting data are not available.

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
