# Peer review of "Recovering from COVID-19 (ReCOV): Feasibility of an Allied-Health-Led Multidisciplinary Outpatient Rehabilitation Service for People with Long COVID"

_ijerph, 2024, doi:10.3390/ijerph21070958_

Round 1

Reviewer 1 Report

Comments and Suggestions for Authors

Methods:

1.     Participants and setting section - The population included was COVID-19 infections persisting for 4 weeks and a potential waiting time was allowed for, but this statement was not reflected in the results. It was not evident to the reviewer that the participants ultimately included in the analysis met the diagnostic criteria for a long COVID (i.e., symptoms persisted for 12 weeks).

2.     Intervention section – “available literature at the time”, suggesting to provide corresponding references.

3.     Outcomes section - This reviewer does not believe that practicality was appropriately assessed in the manuscript, the number of adverse events is a validation of the effectiveness of a particular rehabilitation treatment itself, not the practicality of the ReCOV approach proposed in the manuscript.

Results:

1.     The tables should be a three-line table, and please add a note to the table to adjust the normality of the table.

2.     The final themes that came out of the qualitative research are all not reflected in the results section.

Discussion:

1.     The response rate illustrated in the “Strengths and limitations” section was only 27%, suggesting that additional discussion on adherence should be included.

Comments on the Quality of English Language

English should be improved.

Author Response

3. Point-by-point response to Comments and Suggestions for Authors

Methods

Comment 1: Participants and setting section - The population included was COVID-19 infections persisting for 4 weeks and a potential waiting time was allowed for, but this statement was not reflected in the results. It was not evident to the reviewer that the participants ultimately included in the analysis met the diagnostic criteria for a long COVID (i.e., symptoms persisted for 12 weeks).

Response 1: Thank you for this observation. We agree with this comment and have added further details regarding the time between initial COVID-19 diagnosis and admission to ReCOV

The 143 included participants had a median age of 42.00 [32.00 – 51.00] years,68% (n = 97) were female (Table 3) and were a median of 35 [27 – 48] weeks post initial COVID-19 diagnosis.” [page 7, line 197-198]

Comment 2:  Intervention section – “available literature at the time”, suggesting to provide corresponding references.

Response 2: Thank you, we have added the references

The ReCOV Allied Health led service was modelled on Austin Health’s Long COVID service (another Victorian healthcare provider), available literature at the time [9-13]…” [page 3, line 109.]

Comment 3:  Outcomes section - This reviewer does not believe that practicality was appropriately assessed in the manuscript, the number of adverse events is a validation of the effectiveness of a particular rehabilitation treatment itself, not the practicality of the ReCOV approach proposed in the manuscript.

Response 2: Thank you for this observation. In Bowen’s feasibility framework (which underpins this study), Bowen et al. suggest “positive and negative effects on the target participants” as an example of a Practicality outcome. We believe that adverse events describe a negative effect on participants and therefore have not made any changes.

Results

Comment 1: The tables should be a three-line table, and please add a note to the table to adjust the normality of the table.

Response 1: Thank you, we have updated the tables to reflect three-line format and added a note regarding normality.

Data are presented as median [interquartile range] or count (percentage)” [page 9, line 241]

Comment 1: The final themes that came out of the qualitative research are all not reflected in the results section.

Response 2: We have added “Theme one, two and three” to better indicate our themes.

Thematic analysis of the survey comments identified three themes that are described below. The first theme suggested…” [page 10, line 254 - 255.]

Discussion

Comment 1: The response rate illustrated in the “Strengths and limitations” section was only 27%, suggesting that additional discussion on adherence should be included.

Response 1: Thank you for this comment. As a point of clarification, there were no set number of sessions for attendance and therefore our limited discharge data was not a case of adherence, but rather, “lost to follow up”. We have amended the manuscript to clarify this.

“Integrating the discharge survey into the final ReCOV session may have improved discharge survey completion rates.” [page 14, line 365-366.]

4. Response to Comments on the Quality of English Language

Comment 1: English should be improved.

Response 1: All authors are native English speakers and we have carefully re-read the manuscript to check for any issues.

Reviewer 2 Report

Comments and Suggestions for Authors

Thank you for sharing this interesting manuscript with the scientific community.I report here some general and specific comments about your paper.

(1) ABSTRACT: (1.1) I believe there may be some confusion regarding the "background" section. Please give a short introduction to your study before talking about the objective of the study. (1.2) In the methodology section, it would be beneficial to provide more detailed information on the type of design being used.

(2) INTRODUCTION: More information is needed in this part of the manuscript concerning what is known and what is unknown about long COVID. A more detailed description of this disease would also be beneficial to address this paper. 

(3) METHODS: In order to avoid confusion, it is recommended that the methodology used and specified in the "design" section be more clearly specified. Furthermore, to follow the thread and understand the article, it is important to name each of the parts that will be developed in the results section. 

(3)DISCUSSION: This section should outline the implications for practice and future lines of research. 

Author Response

3. Point-by-point response to Comments and Suggestions for Authors

Abstract

Comment 1: I believe there may be some confusion regarding the "background" section. Please give a short introduction to your study before talking about the objective of the study. (1.2) In the methodology section, it would be beneficial to provide more detailed information on the type of design being used.

Response 1: We have added further details regarding the background and study design, but are limited by the 200-word limit.

Background: A multidisciplinary approach is required for the management of Long COVID. The aim of this study was to determine the feasibility (demand, implementation, practicality, acceptability, and limited efficacy) of an allied health led multidisciplinary, symptom management service (ReCOV) for Long COVID. Methods: A single group, observational cohort feasibility study” [page 1, lines 31-34]

Introduction

Comment 2:  More information is needed in this part of the manuscript concerning what is known and what is unknown about long COVID. A more detailed description of this disease would also be beneficial to address this paper. 

Response 2: We have added additional information into the Introduction.

Over 200 different symptoms of Long COVID have been reported including fatigue, shortness of breath, cough, chest pain, nausea, disordered sleep, memory loss and difficulty concentrating [1, 5]. It has impact on numerous organs and body systems including the heart, blood vessels, lungs, immune system, gastrointestinal tract and neurological system. Hypothesized biological mechanisms for this include neuroinflammation, viral persistence, autoimmunity and excessive blood clotting and more research is required to confirm these hypotheses. … Moreover, as the exact pathophysiological mechanism of Long COVID is not fully understood, there are limited effective treatments but strategies to support symptom management are recommended as fundamental [page 2, line 58-71]

Methods:

Comment 3:  In order to avoid confusion, it is recommended that the methodology used and specified in the "design" section be more clearly specified. Furthermore, to follow the thread and understand the article, it is important to name each of the parts that will be developed in the results section. 

Response 2: Thank you for this observation. We have added further details into the Methods and Results sections as well as naming each of the domains in both the Design section and Outcome sections.

An observational, single-site cohort feasibility study was conducted using a feasibility evaluation framework.” [page 2, line 85.]

“The primary outcomes were the feasibility domains of acceptability, demand, implementation, practicality and limited efficacy.” [page 5, line 135-136.]

“3. Results

Outcomes

Participants and demand” [page 7, line 191.]

Discussion

Comment 1: This section should outline the implications for practice and future lines of research.

Response 1: Implications for practice and future directions for research are described throughout the discussion section. For example, “ensuring that clinicians in the community are adequately equipped to manage this complex condition and have clear referral pathways may also help with ensuring participants’ ongoing needs are met. Finally, balancing the financial sustainability with the above considerations should be incorporated into future undertakings of similar services… Further research is required to determine the clinical and cost effectiveness of such a service, especially with respect to optimal service duration and patient outcomes… Future iterations of the service should ensure the model of care is explained to participants, outcome measures are Long COVID specific and technology platforms and referral pathways are streamlined to improve patient acceptability.” We have provided an additional summary.

Future iterations of a hybrid (telehealth and face-to-face) Long COVID clinic should ensure the model of care is understood by consumers (including the time-limited nature and limited treatment options for Long COVID), technology platforms and appropriate ongoing follow-up are available and outcome measures are Long COVID specific.[page 13, line 315-319]

Round 2

Reviewer 1 Report

Comments and Suggestions for Authors

Thank the author for responding to the question. Thank you to the authors for responding to the question. This reviewer suggests that in addition to adverse events, patient adherence, lost to follow-up rates, etc. might also fit the examples suggested by Bowen et al. for Practicality outcomes, and should they also be included in the references and analyses.

Author Response

Thank you for your prompt response.

Comment 1: This reviewer suggests that in addition to adverse events, patient adherence, lost to follow-up rates, etc. might also fit the examples suggested by Bowen et al. for Practicality outcomes, and should they also be included in the references and analyses.

Response 1: Thank you for this suggestion. To align with Bowen's feasibility framework, we have incorporated the patient adherence (or in this case, commencement to completion) into "Demand" ("actual use" as described by Bowen et al) and the number of sessions completed into "Implementation" ("Degree of execution" as described by Bowen et al.). Lost to follow up rates were already provided "Thirty-nine (27%) completed the discharge limited efficacy testing survey". Both Methods and Results sections have been updated as suggested. 

Demand:

Methods: "Demand for the service was established by recording referral and participant numbers (including demographic details) and completion rates."

Results: "From May to December 2022, 285 people were referred to the ReCOV service and of the 148 that commenced, 143 (97%) completed ReCOV within the study timeframe (Figure 1)."

Implementation:

Methods: "Implementation was evaluated by calculating waitlist times, time from admission to discharge within the service, the number of sessions, and the number of different health professionals accessed within the ReCOV service."

Results: "Despite ReCOV being planned as a 12-week service, participants were seen for a median of 24.00 [16.00 – 34.00] sessions over a median of 17.00 [10.86 – 23.00] weeks."
